# The Early Life Microbiota Is Not a Major Factor Underlying the Susceptibility to Postweaning Diarrhea in Piglets

Martin Beaumont,[a] Corinne Lencina,[a] Allan Bertide,[a] Lise Gallo,[a] Céline Barilly,[a] Christelle Marrauld,[a] Laurent Cauquil,[a] Arnaud Samson,[b] Sylvie Combes[a]

aGenPhySE, Université de Toulouse, INRAE, ENVT, Castanet-Tolosan, France
bADM, Rue de l'Eglise, Château-Thierry Cedex, France

**ABSTRACT** Postweaning diarrhea (PWD) in piglets impair welfare, induce economic losses and lead to overuse of antibiotics. The early life gut microbiota was proposed to contribute to the susceptibility to PWD. The objective of our study was to evaluate in a large cohort of 116 piglets raised in 2 separate farms whether the gut microbiota composition and functions during the suckling period were associated with the later development of PWD. The fecal microbiota and metabolome were analyzed by 16S rRNA gene amplicon sequencing and nuclear magnetic based resonance at postnatal day 13 in male and female piglets. The later development of PWD was recorded for the same animals from weaning (day 21) to day 54. The gut microbiota structure and $\alpha$-diversity during the suckling period were not associated with the later development of PWD. There was no significant difference in the relative abundances of bacterial taxa in suckling piglets that later developed PWD. The predicted functionality of the gut microbiota and the fecal metabolome signature during the suckling period were not linked to the later development of PWD. Trimethylamine was the bacterial metabolite which fecal concentration during the suckling period was the most strongly associated with the later development of PWD. However, experiments in piglet colon organoids showed that trimethylamine did not disrupt epithelial homeostasis and is thus not likely to predispose to PWD through this mechanism. In conclusion, our data suggest that the early life microbiota is not a major factor underlying the susceptibility to PWD in piglets.

**IMPORTANCE** This study shows that the fecal microbiota composition and metabolic activity are similar in suckling piglets (13 days after birth) that either later develop postweaning diarrhea (PWD) or not, which is a major threat for animal welfare that also causes important economic losses and antibiotic treatments in pig production. The aim of this work was to study a large cohort of piglets raised in separates environments, which is a major factor influencing the early life microbiota. One of the main findings is that, although the fecal concentration of trimethylamine in suckling piglets was associated with the later development of PWD, this gut microbiota-derived metabolite did not disrupt the epithelial homeostasis in organoids derived from the pig colon. Overall, this study suggests that the gut microbiota during the suckling period is not a major factor underlying the susceptibility of piglets to PWD.

**KEYWORDS** swine, diarrheal disease, intestinal bacteria, metabolites, organoids, monolayer, epithelium, trimethylamine, butyrate, gut microbiota, metabolomics, pig, weaning, diarrhea

Address correspondence to Martin Beaumont, martin.beaumont@inrae.fr.

The authors declare no conflict of interest.

In piglets, post-weaning diarrhea (PWD) is a major issue since it impairs welfare, causes mortality, induces economical losses, and requires massive utilization of antimicrobials, which contributes to the emergence of resistant pathogens (1–3). PWD is generally observed in piglets around 1 week after weaning and can be caused by multiple factors resulting in an excessive secretion or a reduced absorption of fluids and electrolytes by intestinal epithelial

cells (2, 4, 5). A major trigger of PWD in piglets is an excessive epithelial secretion of ions induced by pathogens such as enterotoxigenic *Escherichia coli* (4, 6, 7). Invasive enteric pathogens such as *Salmonella* can also induce diarrhea through impairment of the intestinal barrier function due to inflammation (2, 6). In early weaned piglets, osmotic diarrhea can also be caused by the accumulation of unabsorbed carbohydrates due to the limited digestive capacity of the immature small intestine (4, 8). Moreover, social stress induced at weaning by the separation from the mother and mixing with new littermates can increase the susceptibility to PWD (2). Weaning also induces a transient intestinal inflammation related to post-weaning anorexia that has also been proposed to predispose pigs to enteric infections (9, 10). Overall, PWD in piglets results from a complex combination of environmental, dietary, and social modifications occurring at weaning.

The gut microbiota could be another important factor underlying the susceptibility to PWD in piglets (3, 11, 12). Indeed, commensal bacteria and their metabolites might play a protective role against diarrheal diseases since they can provide resistance against enteric pathogens, influence intestinal immunity, and enhance epithelial maturation (11, 13, 14). The implication of the gut microbiota in PWD was suggested by the preventive effect of fecal microbiota transplantation from resistant to susceptible piglets (15). This protection was mediated by *Lactobacillus* species enhancing fluid absorption through the production of bacteriocins (15). Interestingly, accelerated maturation of the gut microbiota induced by an early introduction of solid food in suckling piglets also reduced PWD (16). All these observations suggest that specific gut microbiota protect or predispose piglets to PWD. The identification of such a microbial signature would be useful to understand the mechanisms underlying PWD, predict diarrhea development, and design preventive strategies.

Two studies suggested that a specific gut microbiota composition during the suckling period might predispose piglets to PWD (17, 18). However, the 2 studies did not identify a common microbiota signature in suckling piglets that later developed PWD. This inconsistency is probably linked to differences in age at sampling, antimicrobial usage, or limited statistical power. Additionally, the generalization of these results in various contexts (environments and breeding practices) can be questioned. Indeed, the major factors influencing the gut microbiota in suckling piglets are the sow (19, 20) and the rearing environment (21, 22).

In this context, the objective of our study was to evaluate, in a large number of piglets originating from different maternal environments, whether it was possible to detect a specific microbiota signature in suckling piglets that later develop PWD. We also explored the potential contribution of gut microbiota-derived metabolites in the susceptibility to PWD by using a combination of functional predictions, metabolomics, and experiments in pig colon organoids.

## RESULTS

**The early life microbiota is not associated with the later development of PWD.** Our first objective was to determine whether specific gut microbiota signatures during the suckling period were associated with the later development of PWD. We analyzed the microbiota composition by sequencing of 16S rRNA gene amplicons in fecal samples collected at postnatal day 13 from 68 female and 48 male piglets raised in 2 distinct farms ($n = 58$/farm) (Fig. 1A and Table S1). A total of 32% of piglets ($n = 37$ piglets in total; $n = 21$ females and $n = 16$ males representing 31% and 33% of females and males, respectively) had PWD that occurred for the first time mostly between 6 and 10 days after weaning (Fig. 1B and C, and Table S1). Piglets that developed PWD originated from 23 out of 30 litters and tended to be heavier at day 21 (Fig. 1D and Table S1).

The microbiota structure in suckling piglets was not associated with the later development of PWD (PERMANOVA: $P = 0.521$) or with sex ($P = 0.739$) (Fig. 2A). We observed a trend for an interaction between the development of PWD and sex that contributed to a very small fraction of the microbiota structure ($R^2$: 1.0%, $P = 0.062$). The main factors influencing the microbiota structure in suckling piglets were, as expected, the farm ($R^2$: 9.5%, $P < 0.001$) and the sow ($R^2$: 34.2%, $P < 0.001$). There was no significant interaction between PWD and the farm ($P = 0.918$), indicating that the potential links between microbiota and

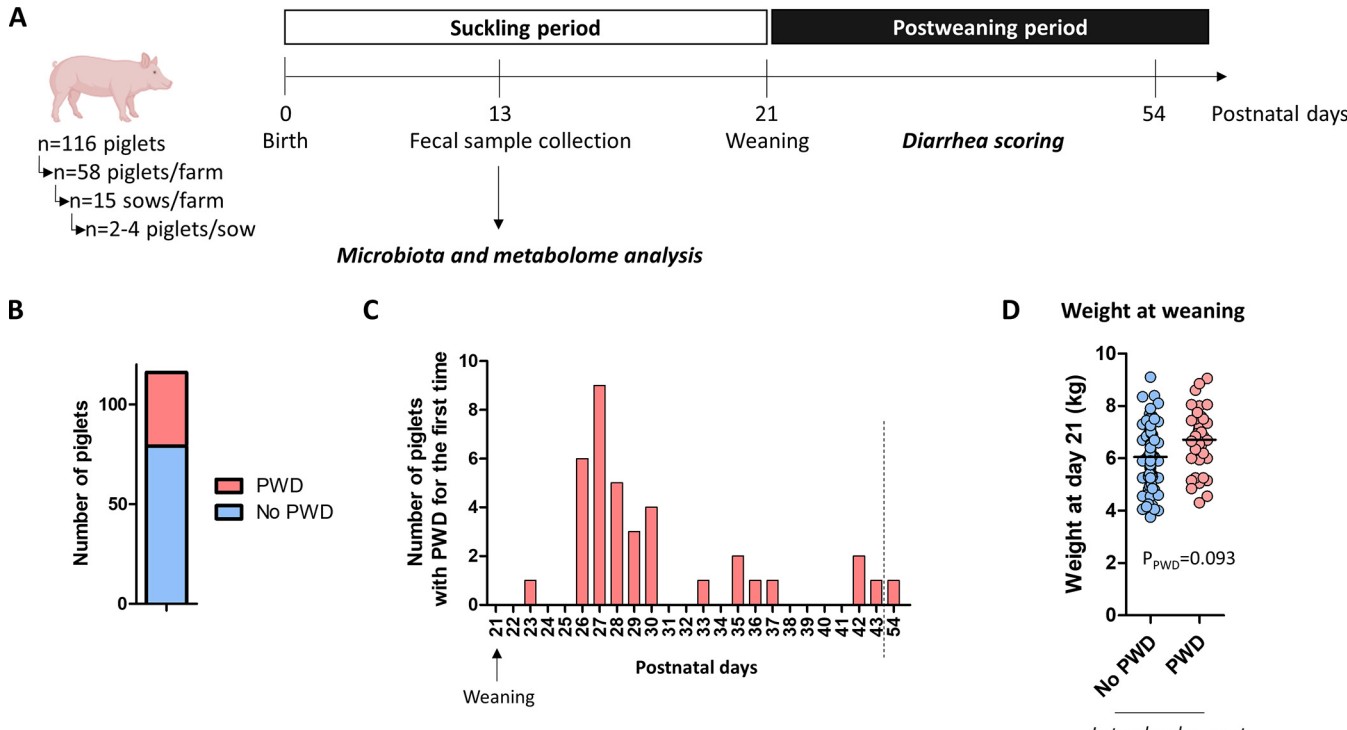

**FIG 1** Experimental design, diarrhea evaluation, and body weight. (A) Schematic representation of the experimental design. Fecal samples were collected at postnatal day 13 from suckling piglets for microbiota and metabolome analysis. Fecal samples were scored for diarrhea after weaning. (B) Number of piglets with or without post-weaning diarrhea (PWD) between postnatal day 21 and 54. (C) Number of piglets with PWD for the first time between postnatal day 21 and 54. (D) Body weight of piglets the day of weaning (postnatal day 21). Dots and bars represent individual and mean values, respectively. Data were analyzed with a linear mixed model (fixed effects: PWD, sex and their interaction, random effects: farm and sow).

PWD were not influenced by the early life environment. The microbiota richness (number of observed OTUs) and Shannon $\alpha$-diversity index in suckling piglets were not associated with the later development of PWD (Fig. 2B). However, we observed that a slightly lower bacterial richness and Shannon index at postnatal day 13 were associated with the later development of PWD in male but not in female piglets (Table S3).

The relative abundance of the 6 dominant phyla (> 0.5%) during the suckling period was not associated with later PWD (Fig. 3A and Table S4). The relative abundance of 24

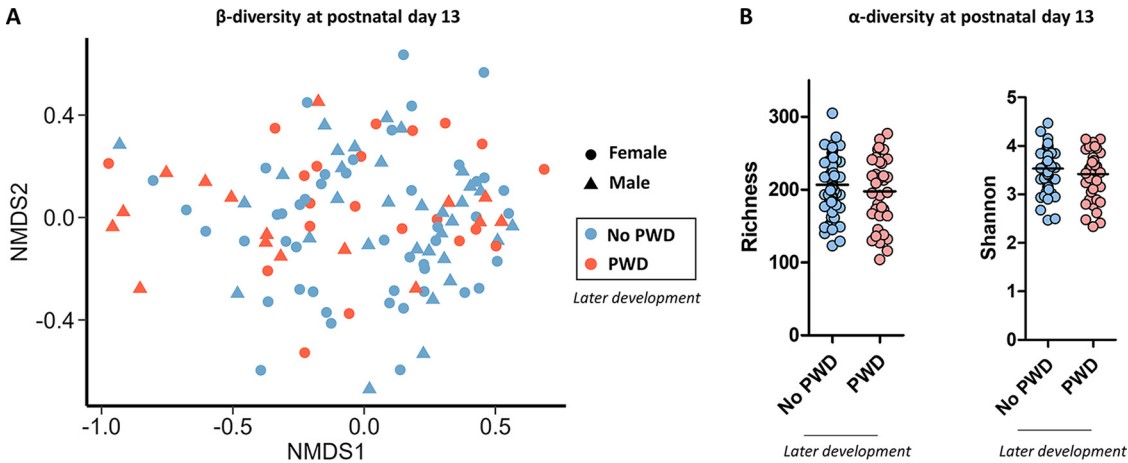

**FIG 2** Microbiota structure and diversity. The gut microbiota was analyzed by 16S rRNA gene amplicon sequencing at postnatal day 13 in suckling piglets that later developed or not postweaning diarrhea (PWD). (A) Non-metric multidimensional scaling (nMDS) representation of the microbiota structure based on the Bray-Curtis distance (Stress = 19.3). (B) Microbiota richness (number of observed OTUs) and Shannon $\alpha$-diversity index. Dots and bars represent individual and mean values, respectively. $\alpha$-diversity data were analyzed with a linear mixed model (fixed effects: PWD, sex and their interaction, random effects: farm and sow).

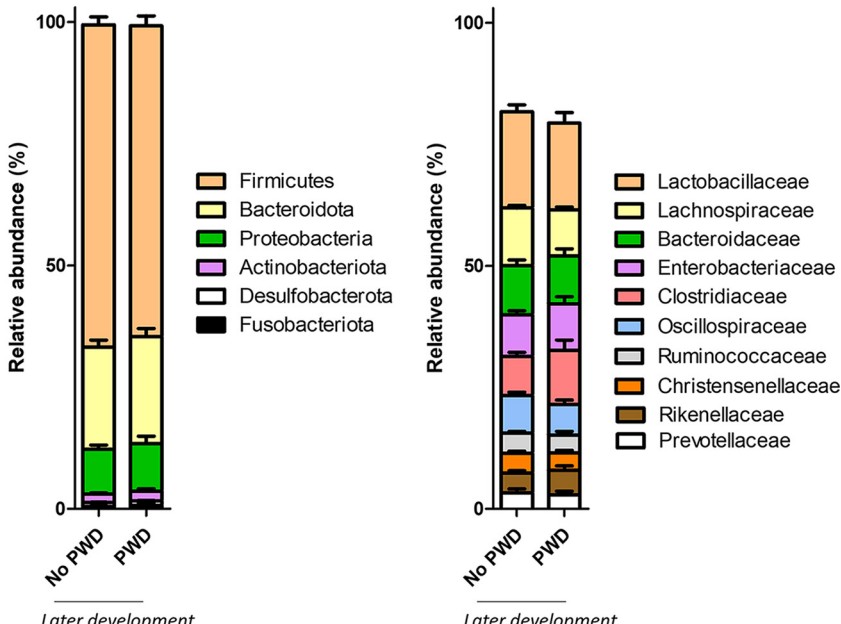

**A  Bacterial phyla at postnatal day 13**

**B  Bacterial families at postnatal day 13**

**FIG 3** Microbiota composition at the phylum and family level. The gut microbiota was analyzed by 16S rRNA gene amplicon sequencing at postnatal day 13 in suckling piglets that later either developed postweaning diarrhea (PWD) or not. (A) Relative abundance of the 6 most abundant bacterial phyla. (B) Relative abundance of the 10 most abundant bacterial families. Bars represent the mean values. Error bars represent the standard error of the mean. Data were analyzed with a linear mixed model (fixed effects: PWD, sex and their interaction, random effects: farm and sow). No significant effect of PWD were observed for the dominant taxa presented in this figure.

dominant bacterial families ($> 0.5\%$) was not different between suckling piglets with or without later PWD (Fig. 3B and Table S5). We observed a few low amplitude and sex-dependent differences in the relative abundances of 5 bacterial families between suckling piglets with or without later PWD (*Christensenellaceae*, *Prevotellaceae*, *Rikenellaceae*, *Oscillospiraceae*, and *p-2534-18B5* gut group) (Table S5). The relative abundance of the 26 dominant bacterial genera ($> 0.5\%$) was not different between suckling piglets with or without later PWD (Table S6). There was a few low amplitude and sex-dependent differences in the relative abundances of 4 bacterial genera between suckling piglets with or without later PWD (*Christensenellaceae R-7 group*, *Rikenellaceae RC9 gut group*, *Prevotellaceae NK3B31 group*, and *NK4A214* group) (Table S6). A PLS-DA model built with the relative abundance of the 26 dominant genera ($> 0.5\%$) at postnatal day 13 failed to predict the later development of PWD (balanced error rate: 50%).

Altogether, our results indicate that there was no global signature of the gut microbiota during the suckling period correlated with the later development of PWD. However, we observed a few unexpected sex-dependent associations between PWD and the microbiota diversity or the abundance of a few bacterial taxa in the early life microbiota. The low amplitude of these changes questions their biological relevance.

**Trimethylamine is the bacterial metabolite which concentration during early life is the most strongly associated with the later development of PWD.** Despite the lack of a taxonomic signature of the early life microbiota associated with the later development of PWD, we hypothesized that there might exist a functional profile of the microbiota linked with the susceptibility to diarrheal disease. Indeed, metabolites produced by the early life microbiota could contribute to PWD through their action on host intestinal cells. Functional inference using PICRUSt2 showed that the relative abundances of 317 predicted metabolic pathways of the gut microbiota in suckling piglets were not associated with the later development of PWD (Table S7). A PLS-DA model built with the relative abundance of

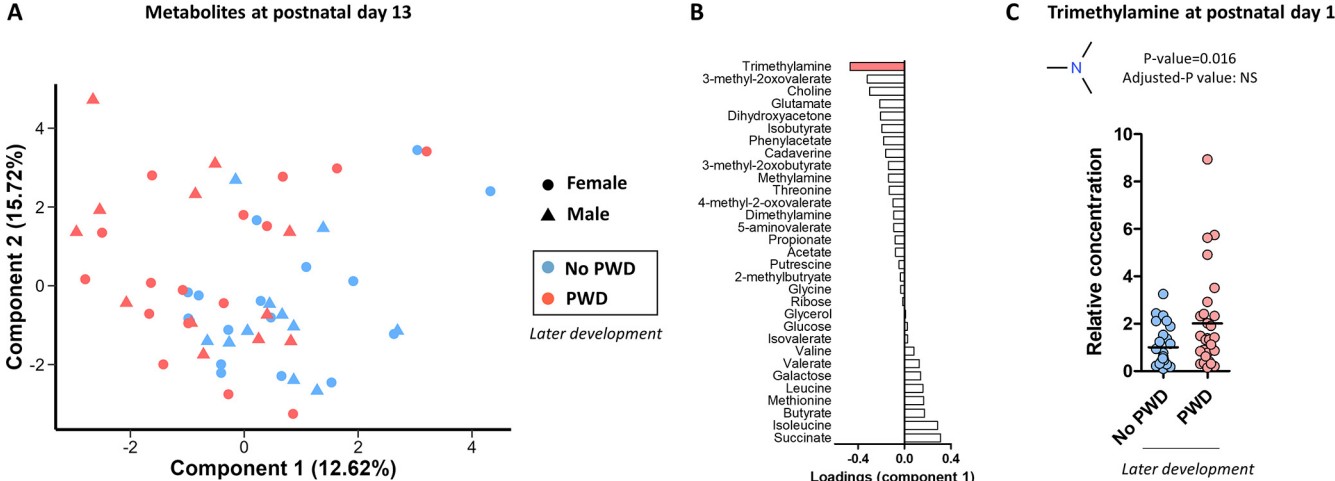

**FIG 4** Fecal metabolome. The fecal metabolome was analyzed by nuclear magnetic resonance-based metabolomics at postnatal day 13 in suckling piglets that later either developed postweaning diarrhea (PWD) or not. (A) Partial-least square discriminant analysis (PLS-DA) built with the relative abundance of 39 metabolites as a data matrix and PWD as a predictor. (B) Loadings of the component 1 of the PLS-DA model. Trimethylamine is highlighted in red since this metabolite contributes the most to the separation between groups. (C) Relative concentration of the bacterial metabolite trimethylamine. Dots and bars represent individual and mean values, respectively. Data were analyzed with a linear mixed model (fixed effects: PWD, random effects: farm and sow).

the 317 predicted metabolic pathways failed to predict the later development of PWD (balanced error rate: 51%).

We then evaluated the actual production of metabolites by the gut microbiota by analyzing the fecal metabolome by using NMR-based metabolomics in a subset of samples ($n = 28$ piglets without PWD and $n = 28$ piglets with PWD). In order to identify the metabolites which concentration during the suckling period was the most strongly associated with the later development of PWD, we used PLS-DA (Fig. 4A). The poor predictive capacity of the PLS-DA model (balanced error rate: 46%) indicated that the global fecal metabolome was not predictive of PWD. Yet, analysis of the PLS-DA loadings on the component 1 (separating the piglets according to later PWD) indicated that trimethylamine (TMA) was the metabolite the most strongly associated with the development of PWD (Fig. 4B). Indeed, the relative concentration of TMA was 2-fold higher in suckling piglets that later developed PWD (Fig. 4C). Among the 39 identified metabolites, TMA was the only one to be significantly associated with later PWD when considering the $P$-value ($P = 0.016$) (Table S8). However, this association was not significant when considering the $P$-value adjusted for multiple testing.

In summary, our data showed that the overall metabolic activity of the early life microbiota was not linked to the later development of PWD. TMA was the bacterial metabolite which concentration during the suckling period was the most strongly associated with the later development of PWD.

**Trimethylamine does not disrupt epithelial homeostasis in piglet colon organoids.** Based on our NMR-based metabolomics results, we hypothesized that an increased exposure to TMA during the suckling period could disrupt intestinal epithelial homeostasis and thereby predispose piglets to PWD. To test this hypothesis, we treated organoids derived from the colon of a suckling piglet with TMA at three concentrations (0.1, 1, and 10 mM) (Fig. 5A). We used butyrate (1 mM) as a positive control since this bacterial metabolite is known to regulate epithelial functions.

TMA had no significant effect on the expression of genes involved in epithelial proliferation, differentiation, enteroendocrine function, absorption, and secretion (Fig. 5B to E). The only exception was a reduced expression of a bile acid-binding protein (fatty acid-binding protein 6 [*FABP6*]) in organoids treated with 10 mM TMA (Fig. 5D). TMA also had no effect on the expression of genes involved in epithelial barrier function, except for a reduction of the expression of mucin 1 gene (*MUC1*) in organoids treated with the highest TMA concentration (Fig. 6B). In contrast, butyrate strongly influenced gene expression in pig colon organoids. Butyrate upregulated the expression of genes involved in lipid metabolism (*FABP1* and *FABP6*), ion transport (sodium-hydrogen

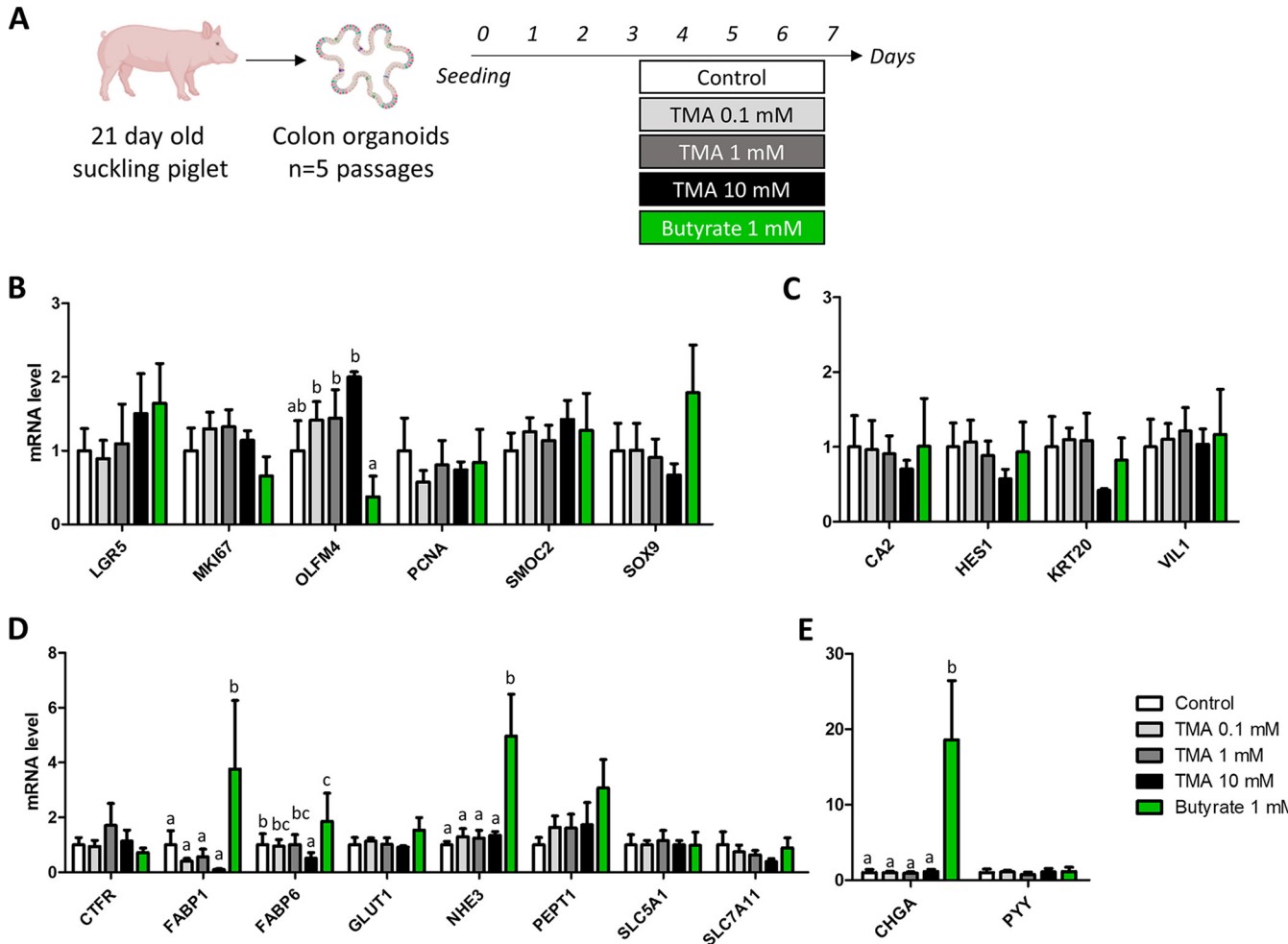

**FIG 5** Effects of trimethylamine and butyrate on gene expression involved in proliferation and differentiation in piglet colon organoids. (A) Schematic representation of the experimental design. Gene expression were analyzed in colon organoids derived from a 21-day old suckling piglet after 4 days of treatment with trimethylamine (TMA) at 3 concentrations (0.1, 1, and 10 mM) or butyrate (1 mM). Relative expression of genes involved in epithelial proliferation (B), differentiation (C), absorption and secretion (D), and enteroendocrine functions (E). Data were analyzed with a linear mixed model (fixed effect: Treatment, random effect: passage). Means labeled without a common letter significantly differ, $P < 0.05$.

exchanger 3 [NHE3]), and enteroendocrine function (chromogranin A [CHGA]), which overall suggest a pro-differentiation effect (Fig. 5C to E). Butyrate also upregulated the expression of genes involved in antimicrobial defenses (lipopolysaccharide binding protein [LBP]; lysozyme [LYZ]; polymeric immunoglobulin receptor [PIGR]) (Fig. 6B, G, and H). Butyrate reduced the gene expression of the mucins MUC1 and MUC2 (Fig. 6B). Interestingly, organoids treated with 10 mM TMA had a higher expression of a proliferation marker (olfactomedin 4 [OLFM4]) when compared to organoids treated with butyrate (Fig. 5B), while the opposite was observed for genes involved in epithelial barrier function (claudin 3 [CLDN3]; mucin 13 [MUC13]; dual oxidase 2 [DUOX2]; chemokine C-C motif ligand 5 [CCL5]) (Fig. 6A, B, D, and E).

As a next step, we evaluated epithelial morphology and barrier function in colon organoid cell monolayers treated at the apical side with TMA (1 mM) or butyrate (1 mM) for 2 days. TMA did not alter the morphology of the cell monolayer, nor the expression pattern of the tight junction protein occludin and did not modify TEER (Fig. 7A to D). In contrast, butyrate enhanced TEER and this effect seemed to be associated with a stronger staining of occludin at cell junctions. Overall, our data obtained in pig colon organoids indicate that, in contrast with butyrate, TMA had very limited effects on epithelial homeostasis. These results suggest that TMA produced by the microbiota of suckling piglets does not predispose to PWD through a disruption of epithelial homeostasis.

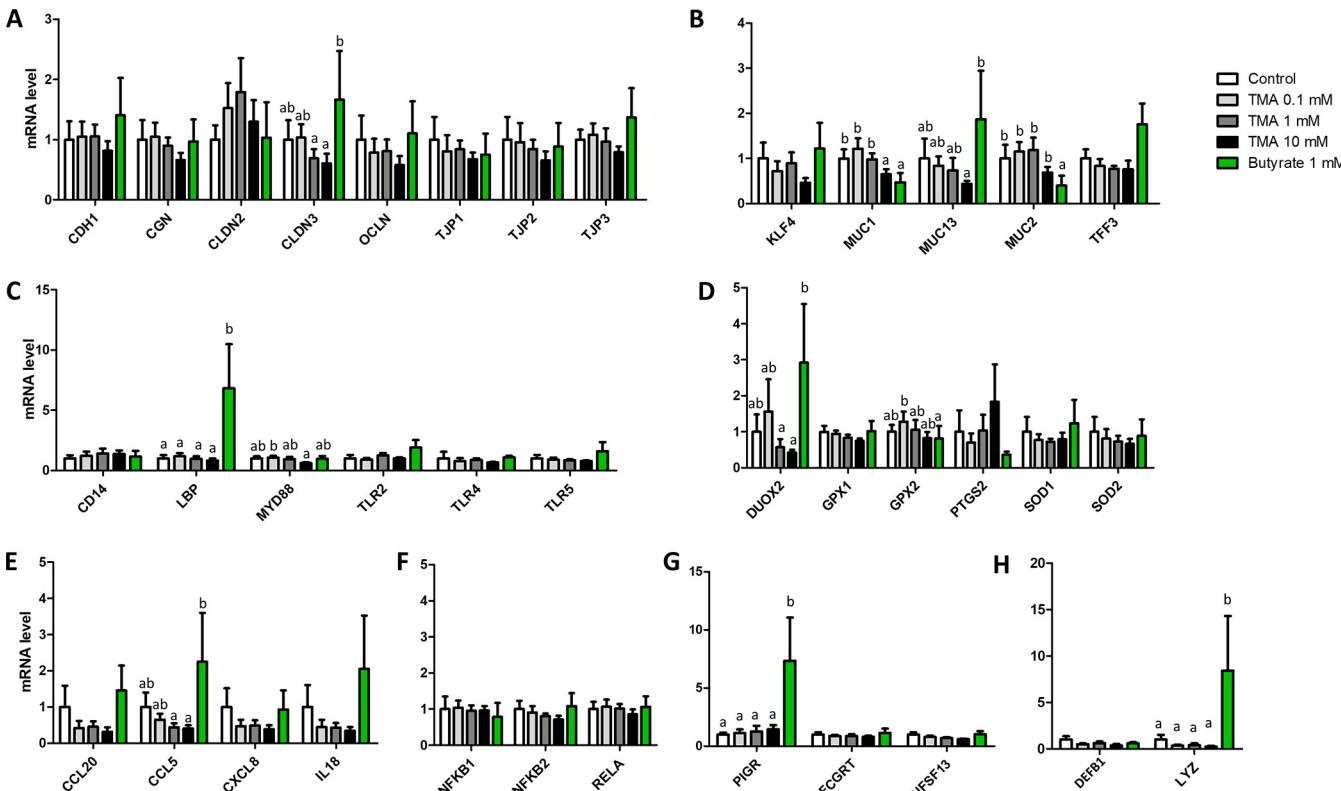

**FIG 6** Effects of trimethylamine and butyrate on gene expression involved in epithelial barrier function in piglet colon organoids. Gene expression were analyzed in colon organoids derived from a 21-day old suckling piglet after 4 days of treatment with trimethylamine (TMA) at 3 concentrations (0.1, 1, and 10 mM) or butyrate (1 mM). Relative expression of genes coding for proteins involved in tight junction formation (A), mucus production (B), Toll-like receptor signaling (C) redox regulations (D), cytokine signaling (E), NF-KB signaling (F), immunoglobulin transport (G), and antimicrobial peptides (H). Means labeled without a common letter significantly differ, $P < 0.05$.

## DISCUSSION

The gut microbiota composition in suckling piglets is strongly influenced by the maternal environment due to colonization by bacteria derived from the mother and from housing, under the influence of selection factors provided by maternal milk and by piglet intrinsic intestinal characteristics (19–23). Accordingly, we observed in our study that the sow and the farm were the major factors shaping the gut microbiota structure in suckling piglets at postnatal day 13. In contrast, we observed no link between the overall gut microbiota structure in suckling piglets and the later development of PWD, which is in agreement with previous results obtained 3 days before weaning in piglets that later developed PWD (18). Other studies on calves, rhesus macaque, and human infants similarly showed that the microbiota structure in early life was not associated with the later development of diarrheal diseases (24–26). The susceptibility to PWD could be mediated by a high abundance in early life of pathobionts involved in diarrheal diseases (e.g., *Escherichia* and *Salmonella*) or by a depletion of protective taxa (e.g., *Lactobacillaceae*) (3, 4, 9, 15). However, we found no significant association between the relative abundance of *Enterobacteriaceae* or *Lactobacillaceae* in suckling piglets and the later development of PWD.

Unexpectedly, we observed a few sex-dependent associations between the gut microbiota diversity in suckling piglets and the later development of PWD. Sexual dimorphism in intestinal microbiota composition and diversity has been described previously in young mammals and are thought to be related to early immune or hormonal differences between males and females (27). The slightly lower microbiota diversity that we observed only in male suckling piglets that later developed PWD is in agreement with a previous study showing a lower diversity in the gut microbiota of 7-day old suckling piglets with later PWD (17). A lower microbiota diversity could contribute to infection by diarrheal pathogens due to a reduced colonization resistance provided by commensal bacteria (13, 28). However, a reduced

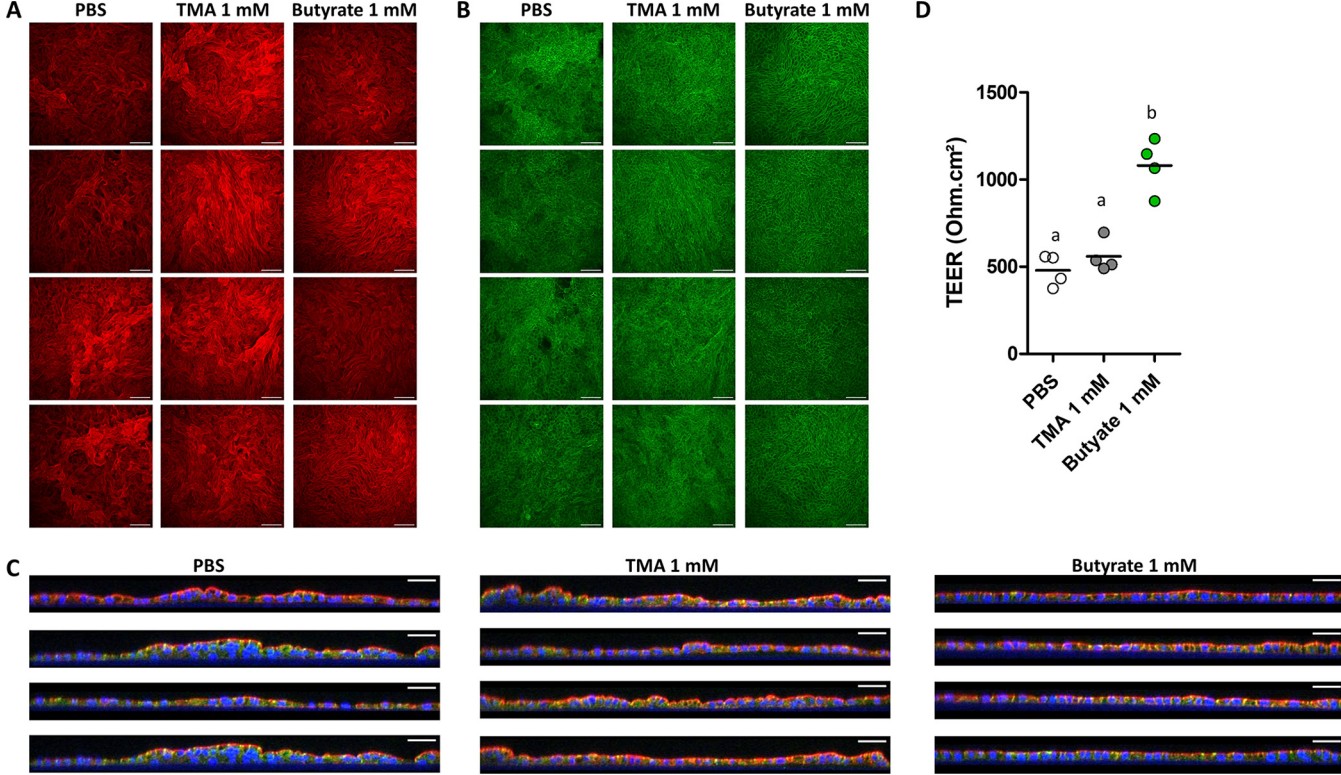

**FIG 7** Effects of trimethylamine and butyrate on epithelial barrier function in cell monolayers derived from piglet colon organoids. Organoid cell monolayers derived from a suckling piglet were treated at the apical side with trimethylamine (TMA), butyrate (1 mM), or PBS (negative control) for 2 days. (A and B) Confocal microscopy images (XY maximum intensity) of organoid cell monolayers. Actin, red; Occludin, green. Scale bar = 100 $\mu$m. (C) Confocal microscopy images (XZ projections) of organoid cell monolayers. Actin, red; Occludin, green; DNA, blue. Scale bar = 50 $\mu$m. (A to C): each image in a panel corresponds to a single cell culture insert. (D) Transepithelial electrical resistance (TEER). Means labeled without a common letter significantly differ, $P < 0.05$.

microbiota diversity is not a necessary condition for the development of PWD since we found that the microbiota diversity in female suckling piglets was not linked to the later development of diarrheal disease, as reported in other species (24–26).

Overall, our data showing no overall association between microbiota features in suckling piglets and PWD suggest that the contribution of the early life microbiota to diarrheal disease is probably low. Our results also imply that 16S rRNA amplicon profiling of the gut microbiota of suckling piglets is not likely to be suitable to identify predictive markers of PWD. Interestingly, Rhoades et al. demonstrated that metagenomic sequencing but not 16S rRNA amplicon profiling was able to identify bacterial taxa associated with the susceptibility to diarrheal diseases, suggesting that a strain-resolution might be required (24). The lack of identification of a clear microbiota signature in suckling piglets with later PWD could also be linked to the multiple possible causes of diarrhea after weaning (osmotic, secretory, and inflammatory) (2). We cannot exclude that a gut microbiota signature might exist in some subtype of PWD and the lack of identifying the cause of diarrhea is a limitation of our study (e.g., by screening the presence of pathogens in the feces of piglets with PWD). Moreover, we used a binary scoring of PWD (yes/no) to ensure a clear identification of piglets that remained healthy after weaning but this method did not allow us to distinguish piglets with mild or severe diarrhea according to the duration of symptoms. The large time scale of PWD development in our study (between postnatal day 23 and 54) might also induce variability due to different mechanisms potentially involved in early and late PWD. However, we did not perform sub-class analysis in order to keep a high statistical power ($n = 37$ piglets with PWD) which might have been lacking in previous studies (17). Moreover, examination of the microbiota $\beta$-diversity did not reveal clustering according to the early or late occurrence of post-weaning diarrhea (data not shown).

The timing of sampling for microbiota analysis might also be a key determinant of the capacity to predict the later development of PWD. Indeed, we analyzed the microbiota at

postnatal day 13 while other studies in piglets focused on earlier (day 7) or later (day 27) time points (17, 18). On the one hand, earlier time points might not be optimal to predict later PWD since a high-interindividual variability is observed in the primocolonizing microbiota, which is dynamic and strongly influenced by stochastic events due to low microbial density (29, 30). On the other hand, the timing and order of bacterial taxa primocolonization has long-standing consequences for the microbiota structure and health of the animal host (30, 31). In contrast, the lower inter-individual variability of microbiota at later time point might increase prediction capacity and also potentially allow the detection of diarrheal pathogens.

A previous study found that minimal taxonomic differences between the microbiota of infant monkeys, whether associated with the later development of diarrhea, translated into functional differences, notably related to the capacity to produce short chain fatty acids that promote gut health (24, 32). In our study, the predicted metabolic capacity of the gut microbiota in suckling piglets and the fecal concentrations of short chain fatty acids (acetate, propionate, and butyrate) were not associated with the later development of PWD. TMA was the only metabolite that fecal concentration in suckling piglets was associated with later PWD. This metabolite is produced by intestinal bacteria from several substrates, including choline which is an essential nutrient provided in large amounts in sow milk (33, 34). Choline is degraded by the bacterial enzyme CutC, which is present in a wide diversity of bacterial genera including some pathobionts (e.g., *Escherichia*, *Shigella*, and *Desulfovibrio*) (34). The trend for a higher body weight for weaning piglets that later developed PWD might suggest a higher level of milk intake and thus potentially greater supply of choline to the gut microbiota for the production of TMA. Thus, we hypothesized that an increase production of TMA by the microbiota during the suckling period might weaken the gut barrier and therefore predisposed piglets to PWD.

Previous experiments in the Caco-2 human cell line showed that TMA reduced cell growth but did not alter the epithelial barrier function (35, 36). To circumvent the lack of cellular diversity and genomic abnormalities of cell lines, we tested the effects of TMA in suckling piglet colon organoids that contain the main cell types and reproduce the 3D architecture of the intestinal epithelium (37, 38). Our results showed that, contrary to our hypothesis, TMA did not alter epithelial homeostasis when considering cell morphology, TEER, and the gene expression of proteins involved in tight-junctions, epithelium renewal, innate immunity, redox enzymes, or ion transport. In contrast, in the same organoid model, we observed that the gut microbiota-derived metabolite butyrate improved epithelial barrier function and enhanced differentiation and the expression of key innate immune factors, which is in agreement with numerous studies demonstrating the protective effect of butyrate on the epithelial barrier (32, 39) and thus validating our experimental model. Based on our results obtained in organoids that contain only epithelial cells, TMA is unlikely to promote PWD in piglets through a disruption of the intestinal epithelial barrier. *In vitro* experiments in human lymphocytes showed that TMA reduced the secretion of the pro-inflammatory cytokines TNF-$\alpha$ and IL1$\beta$ (36). Thus, TMA could potentially predispose piglets to PWD through immune dysregulation. TMA could also be linked to PWD via the hepatic production of trimethylamine-N-oxide (TMAO), which induce pro-inflammatory responses at the systemic level (34, 40). These hypotheses should be tested in future experiments.

**Conclusion.** Our data show that, in a large number of piglets raised in separate environments, there was no specific microbiota signature during the suckling period associated with the later development of PWD. Thus, the early life microbiota is probably not a major factor underlying the susceptibility to PWD in piglets, at least when considering all possible etiologies. Our results also indicate that 16S rRNA gene amplicon sequencing in the fecal samples of suckling piglets is not a suitable method to predict the later development of PWD. TMA was the gut microbiota metabolite that was most prevalent in fecal matter during the suckling period and was the most strongly associated with the later development of PWD. However, our experiments in organoids do not support a direct causal role of TMA in the predisposition to PWD through disruption of epithelial homeostasis.

## MATERIALS AND METHODS

**Animals and sample collection.** A total of 116 Piglets (Piétrain x Large white x Landrace) raised in 2 farms ($n$ = 58 piglets/farm, $n$ = 34 females and $n$ = 24 males in each farm) were studied (Fig. 1A). Between 2 and 4 piglets from each sow were studied ($n$ = 30 sows in total, $n$ = 15 sows per farm). The selected piglets were healthy and representative of the mean weight of their litter. Primiparous and multiparous sows were included in the study. The sows and the piglets were fed following the usual procedures of each farm, as described previously (21). All piglets were weaned at postnatal day 21 and moved into the same post-weaning farm. Weaned piglets from the 2 farms were not mixed in the same pens. Two post-weaning rooms contained only piglets from 1 farm ($n$ = 37 in room with piglets from farm 1 and $n$ = 40 in the room with piglets from farm 2) and the third room contained piglets originating from both farms ($n$ = 21 piglets from farm 1 and $n$ = 18 from farm 2). Diarrhea were recorded daily from weaning (day 21) until day 54 by scoring fecal samples (0= no diarrhea, 1 = diarrhea). Piglets were individually classified as diarrheic when diarrhea was observed at least once during the post-weaning period. None of the piglets received antibiotic treatment. Fecal samples were collected once at postnatal day 13 in healthy piglets (i.e., during the suckling period) and stored at $-80°C$ until analysis.

**16S rRNA gene amplicon sequencing and sequence analysis.** DNA was extracted from 50 mg of fecal samples using the Quick-DNA Fecal/Soil Microbe 96 Kit (ZymoResearch) according to the manufacturer's instructions. PCR amplicons of the 16S rRNA gene V3-V4 region were sequenced by MiSeq technology (Illumina) at the Genomic and transcriptomic platform (GeT-PlaGe, INRAE), as described before (41). Sequencing reads were deposited in the National Center for Biotechnology Center for Biotechnology Information Sequence (accession number: PRJNA591810 and PRJNA904234, sample list and corresponding metadata are presented in Table S1). Amplicon sequences were analyzed by the FROGS pipeline version 3.2.3 (42), following the guidelines. Merged sequences were selected based on their size (350 to 500 nucleotides), dereplicated, and counted. Sequences were then clustered into OTUs using Swarm (clustering aggregation distance: 1). After PCR chimera removal, the OTUs present in less than 3 samples or whose proportion represented less than 0.005% of all sequences were filtered out. The taxonomic affiliation of OTUs was performed with the 16S SILVA database (138.1, pintail 100). A phyloseq object with OTU count table and sample metadata was created. The mean number of reads per sample was 16,181 (min: 7,120 to max: 34,354). For $\alpha$ and $\beta$-diversity analyses, the count table was rarefied to 7,120 sequences per samples with the R software (4.2.0) and the phyloseq package (1.40.0). Microbiota richness (number of observed OTUs) and Shannon $\alpha$-diversity index were calculated. $\beta$-diversity was analyzed with the distance of Bray-Curtis and visualized by non-metric multidimensional scaling (nMDS). The unrarefied count table was used to calculate the relative abundances of bacterial taxa at the phylum, family, and genus level. The functional potential of the gut microbiota was predicted with PICRUSt2 (43) as implemented in FROGS (functional analysis tools) and according to the guidelines. OTUs were placed in the PICRUSt2 reference tree with epa-ng with a minimum alignment length of 80%. Hidden state prediction was performed with the maximum parsimony method with MetaCyc EC-Numbers. OTUs with a nearest sequenced taxon index (NSTI) $>$ 0.2 were excluded from analysis to improve accuracy of prediction. Thus, 429 OTUs representing 71% of the total number of sequences were used for functional predictions. For each sample, the abundances of MetaCyc pathways were calculated and normalized by the total abundance of all pathways.

**Nuclear magnetic resonance-based metabolomics.** The fecal metabolome was analyzed by $^1$H nuclear magnetic resonance (NMR)-based metabolomics as described previously (21) in a subset of piglets ($n$ = 28 piglet without PWD and $n$ = 28 piglets with PWD, originating equally from the 2 farms and with a similar proportion of females and males). Briefly, the metabolites were extracted from fecal samples (100 mg) by homogenization in a buffer prepared in $D_2O$ with a FastPrep instrument (MP Biomedicals) and by successive centrifugation cycles. The spectra were acquired at 300 K using the Carr-Purcell-Meiboom-Gill spin-echo pulse sequence with presaturation on a AVANCE III HD NMR spectrometer operating at 600.13 MHz for $^1$H resonance frequency using a 5 mm inverse detection CryoProbe (Bruker) at the metabolomics platform MetaToul-AXIOM (INRAE). Raw spectra were processed with the galaxy tool Workflow4Metabolomics and metabolites were identified by comparison with the spectra of pure compounds, as described previously (21). A total of 39 metabolites were identified and quantified by using the area under the curve (AUC) of a 0.01 ppm bin (bucket) corresponding to 1 peak of the targeted metabolite non-overlapping with other signals. The relative concentrations of metabolites were calculated by dividing the AUC measured in each sample by the mean of AUC measured in piglets without PWD.

**Experiments with colon organoids.** We used colon organoids derived from a 21-day-old suckling piglet and cryopreserved at passage 1, as described previously (37). A cryovial containing colon organoids was thawed at 37°C and centrifuged (at 300 g, 5 min, and room temperature). The supernatant was removed and the organoid pellet was resuspended in ice-cold Matrigel (Corning) and seeded in pre-warmed 48-well plates (25 $\mu$L drops). After incubation for polymerization (at 37°C, 5% CO2, and 20 min), organoids were cultured with IntestiCult Organoid growth Medium (Human) (Stem cell technologies) supplemented with 1% penicillin/streptomycin. The medium was replaced every 2 to 3 days. Every week, after washing with warm PBS, organoids were broken by pipetting in EDTA-trypsin 0.25% wt/vol (Thermo Fisher Scientific) before incubation (at 37°C, 5% $CO_2$, and 10 min). After centrifugation (at 300 g, room temperature, and 5 min), the organoid cell pellet was re-seeded in Matrigel with a dilution ratio 1:6 and cultured as described above. At day 3 post-seeding, organoids were treated with trimethylamine (Sigma-Aldrich) at 3 concentrations (0.1, 1, or 10 mM), with 1 mM butyrate as a positive control (Sigma-Aldrich), or IntestiCult medium alone as a negative control. The media with or without treatments were refreshed on day 5. At day 7, organoids (6 wells/treatments) were harvested in TRI Reagent (ZymoResearch) and frozen at $-80°C$ until RNA extraction. Experiments were repeated five times (passages 3 to 8). A total of 500 ng RNA purified with Direct-zol RNA MiniPrep Plus kit (Zymo Research) were retrotranscribed into cDNA with the kit GoScript Reverse Transcription Mix, Random primer (Promega) as described previously (37). Gene expression was analyzed by real-time qPCR using Biomark microfluidic system using a Dynamic Array IFC for gene expression (Fluidigm) according to the manufacturer recommendations

and with specific primers (Table S2). The genes which expression was analyzed were selected to characterize the intestinal epithelium renewal and differentiation and to evaluate the barrier function. Expression values were calculated with the $2^{-\Delta Ct}$ method with *GAPDH* gene expression used as a reference. For graphical representations, expression values were divided by the expression value in the negative control group at the same passage. Raw expression values were used for statistical analysis.

Cell monolayers derived from colon organoids of a suckling piglet were obtained as described previously (44). Briefly, $2.5 \times 10^5$ cells were seeded in cell culture inserts (Corning) previously coated with human collagen IV (Sigma-Aldrich). The culture medium was IntestiCult supplemented with 20% FBS (Thermofisher) and 10 $\mu$M Y27632 (ATCC). After 24 h (post-seeding), the basal medium was replaced by IntestiCult supplemented with 20% FBS and cell monolayers were treated at the apical side with trimethylamine (TMA; 1 mM), butyrate (1 mM), or PBS (negative control). Treatments were prepared in PBS. Transepithelial resistance was measured with a cellZscope + (nanoAnalytics), according to the manufacturer's instructions. Treatments were refreshed after 24 h. Two days after treatment, cell monolayers were used for staining of occludin and actin, following a procedure described previously (44).

**Statistical analysis.** The R software (4.2.0) was used for statistical analyses. All data were analyzed by linear mixed models (lme4 and car packages). For microbiota and metabolomics data, fixed effects included the later development of PWD (no or yes), sex (female or male), and their interaction (PWD x sex). Random effects included the farm (maternity farm 1 or 2) and the sow, nested in the farm. *P*-values were corrected for multiple testing using the Benjamini-Hochberg method. Statistical analyses were performed only for bacterial taxa with relative abundance over 0.5% within at least 1 group to ensure reproducibility of the quantification (45). The relative abundance of bacterial taxa and of predicted pathways were transformed to the power of 0.25 before analysis. The relative concentrations of metabolites were $\log_{10}$ transformed before analysis. When the interaction between PWD and sex was significant, least square means were compared between piglets without or with PWD within each sex (emmeans package). PERMANOVA with 999 permutations was used to study the effects of PWD, sex, farm, sow, and interactions between the microbiota structure (vegan package). Partial-least square discriminant analysis (PLS-DA) models were used to evaluate the ability to predict the later development of diarrhea based on the relative abundance of bacterial genera, predicted pathways, or the relative concentration of metabolites during the suckling period (mixOmics package). Data obtained in organoids were analyzed with mixed models as described above with treatment as a fixed effect and passage as a random effect. Relative gene expression values were $\log_{10}$ transformed before analysis.

## SUPPLEMENTAL MATERIAL

Supplemental material is available online only.

**SUPPLEMENTAL FILE 1**, XLSX file, 0.1 MB.

**SUPPLEMENTAL FILE 2**, PDF file, 0.1 MB.

## ACKNOWLEDGMENTS

We are grateful to the genotoul bioinformatics platform Toulouse Occitanie (Bioinfo Genotoul, https://doi.org/10.15454/1.5572369328961167E12) for providing computing and storage resources. We also thank the GenoToul platforms for metabolomics (MetaToul-Axiom) and genomics and transcriptomics (GeT-PlaGE).

Martin Beaumont conceptualized, formally analyzed the data, performed the investigation, and wrote the original draft. Corinne Lencina, Lise Gallo, Céline Barilly, and Christelle Marrauld performed the investigation. Allan Bertide and Laurent Cauquil formally analyzed the data: Arnaud Samson conceptualized, wrote, reviewed, and edited the study. Sylvie Combes Conceptualized, formally analyzed the data, performed the investigation, wrote, – reviewed, and edited the manuscript, and acquired the funding.

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
