## [Reviewer comments · Microbiology Spectrum]

Microbiology Spectrum

The early life microbiota is not a major factor underlying the susceptibility to postweaning diarrhea in piglets

Martin Beaumont, Corinne Lencina, Allan Bertide, Lise Gallo, Céline Barilly, Christelle Marraud, Laurent Cauquil, Arnaud Samson, and Sylvie COMBES

Corresponding Author(s): Martin Beaumont, Institut national de recherche pour l'agriculture l'alimentation et l'environnement

Review Timeline:

Submission Date:	February 15, 2023
Editorial Decision:	April 2, 2023
Revision Received:	June 2, 2023
Accepted:	June 9, 2023

Editor: Maristela Camargo

Reviewer(s): Disclosure of reviewer identity is with reference to reviewer comments included in decision letter(s). The following individuals involved in review of your submission have agreed to reveal their identity: Wen Ren (Reviewer #2)

Transaction Report:

DOI: <https://doi.org/10.1128/spectrum.00694-23>

April 2, 2023

Dr. Martin Beaumont
Institut national de recherche pour l'agriculture l'alimentation et l'environnement
Castanet Tolosan
France

Re: Spectrum00694-23 (The early life microbiota is not a major factor underlying the susceptibility to postweaning diarrhea in piglets)

Dear Dr. Martin Beaumont:

Link Not Available

Sincerely,

Maristela Camargo

Journals Department
Reviewer comments:

Reviewer #1 (Comments for the Author):

Beaumont et al. used 16S rRNA gene amplicon sequencing and nuclear magnetic based resonance aimed to evaluate in a large cohort of 116 piglets raised in two separate farms whether the gut microbiota composition and functions during the suckling period were associated with the later development of PWD. Overall, the topic is interesting and important for the livestock field. However, I have several concerns that should be addressed.

I am concerned about the novelty of the manuscript. While the authors mention in the manuscript that previous studies have found that microbiota is important for the susceptibility to postweaning diarrhea in piglets, they also state that in the present study, microbiota is not a major factor and attribute this to the sow and rearing environment. As there are already many studies

focused on the sow and rearing environment, the authors should provide more explanation about the novelty of their study.

The authors conclude that gut microbiota is not a major factor in postweaning diarrhea in piglets based solely on 16S rRNA gene amplicon sequencing. To increase confidence in their conclusion, I suggest the authors provide more data, such as using fecal microbiota transplantation in mice to determine if the two groups could or could not increase the susceptibility to postweaning diarrhea.

I am curious about the justification for the use of 116 piglets. Did the authors undertake a power calculation to ensure this number was sufficient for their study?

Figure 3 shows that the total family levels do not reach 100%. I would like to know the reason for this discrepancy.

The authors aim to find the effects of trimethylamine and butyrate on gene expression involved in proliferation and differentiation in piglet colon organoids using qPCR. However, it is not acceptable to rely solely on qPCR data, and the authors should also determine the protein levels to provide a more complete picture.

Reviewer #2 (Comments for the Author):

Line 61-62: the formal of reference is not correct;

Line 99-100: please make it clear the number of piglets in each room;

Line 100-103: although there are different diarrhea scoring systems, all of them include light diarrhea and severe diarrhea, the difference on diarrhea may influence the microbiota a lot. However, current study only record diarrhea or not, and even include piglets only observed at once (these piglets may even not show microbiota changes), I will treat this as a big limitation of this study, this may cause the current results that no difference between normal and diarrhea piglets. If I can suggest, please use diarrhea samples that piglet show severe diarrhea in at least continue 3 days;

Line 101-102: as far as I understand, feces samples will be collect at a very long period (1 month period: d21 to d 54), this means some samples may collected from d 22, dome samples may collected from d 50, this is the second shortage of this study: long time period will influence the results a lot! It means this will cause high variation to the results in the same group.

Line 104-105: not clear enough to show sample collection. In current description: "Fecal samples were collected during the suckling period at postnatal day 13 in healthy piglets...", do this mean only 1 time point of feces sample was collect?

Line 176-179: please make it clear in here what are the genes you have analyzed in this study.

Line 255 & 285: Please aware this is the results part, not discussion and reference should in this part.

Line 280-285: this should be in the method part other than in the results part;

Staff Comments:

Preparing Revision Guidelines

Please return the manuscript within 60 days; if you cannot complete the modification within this time period, please contact me. If you do not wish to modify the manuscript and prefer to submit it to another journal, please notify me of your decision immediately so that the manuscript may be formally withdrawn from consideration by Microbiology Spectrum.

Spectrum00694-23 (The early life microbiota is not a major factor underlying the susceptibility to postweaning diarrhea in piglets)

Dear editor,

We thank the reviewers for their time in reviewing our manuscript. Please find below our point-by-point responses to each of the comments as well as a description of modifications of the manuscript. Line numbers refer to the revised version of the manuscript with modifications marked. We also have modified the formatting of the manuscript according to guidelines. We hope that these modifications lead to the acceptance of this revised manuscript to *Microbiology Spectrum*. Thank you for your consideration,

Martin Beaumont, corresponding author

Reviewer #1 (Comments for the Author):

Beaumont et al. used 16S rRNA gene amplicon sequencing and nuclear magnetic based resonance aimed to evaluate in a large cohort of 116 piglets raised in two separate farms whether the gut microbiota composition and functions during the suckling period were associated with the later development of PWD. Overall, the topic is interesting and important for the livestock field. However, I have several concerns that should be addressed.

I am concerned about the novelty of the manuscript. While the authors mention in the manuscript that previous studies have found that microbiota is important for the susceptibility to postweaning diarrhea in piglets, they also state that in the present study, microbiota is not a major factor and attribute this to the sow and rearing environment. As there are already many studies focused on the sow and rearing environment, the authors should provide more explanation about the novelty of their study.

We found only two previous studies that explored the potential link between the early life microbiota and postweaning diarrhea in piglets (Dou et al, Plos One, 2018, PMID: 28072880 ; Karasova et al., Res Vet Sci, 2021, PMID: 33444908). However, each study identified different bacterial taxa associated with the predisposition to postweaning diarrhea. This inconsistency could be related to the important weaknesses of these studies including a low statistical power (n=5 piglets/group in Dou et al.) and antibiotic usage in Karasova et al, as indicated in the introduction (lines 84-90). Based solely on these two publications, we consider that the scientific evidence is not sufficient to conclude whether the early life microbiota could predispose piglets to postweaning diarrhea. Moreover, the results of these two previous studies were obtained in the context of a single rearing environment, which has a major influence on the gut microbiota, as mentioned by the reviewer. Thus, the aim of our study was to overcome these limitations by analyzing the gut microbiota in a large number of suckling piglets (n=116) raised in two separate environments and originating from 30 litters. This experimental design provided strong statistical power and, in these conditions, we found no association between the gut microbiota of suckling piglets and the latter development of postweaning diarrhea. Thus, the novelty of our study is to provide strong experimental data showing that the early life microbiota is not a major factor driving postweaning diarrhea. Additionally, we analyzed the metabolic activity of the gut microbiota, which was not addressed in previous studies focusing on the link between the early life microbiota and postweaning diarrhea. This approach allowed us to identify that the fecal concentration of trimethylamine in suckling piglets was associated with the latter development of post-weaning diarrhea. This result is novel and we also explored for the first time the effects of this metabolite on the pig intestinal epithelium by using organoids.

The authors conclude that gut microbiota is not a major factor in postweaning diarrhea in piglets based solely on 16S rRNA gene amplicon sequencing. To increase confidence in their conclusion, I suggest

the authors provide more data, such as using fecal microbiota transplantation in mice to determine if the two groups could or could not increase the susceptibility to postweaning diarrhea.

We agree with the reviewer that microbiota transplantation experiments is a powerful approach to demonstrate that a specific gut microbiota is involved in a specific phenotype, such as postweaning diarrhea in the current study. However, we did not identify a specific microbial community associated with the latter development of postweaning diarrhea. Thus, we consider that the rationale for transferring the microbiota of suckling piglets with or without postweaning diarrhea to mice is low. Moreover, our study is not solely based on 16S rRNA gene amplicon sequencing since we also analyzed the fecal metabolome by using nuclear magnetic resonance. We also combined these *in vivo* experiments with *in vitro* experiments in pig intestinal organoids.

I am curious about the justification for the use of 116 piglets. Did the authors undertake a power calculation to ensure this number was sufficient for their study?

Power calculation shows that for an effect size of 5 % and a standard error of 7%, which realistic for the relative abundance of bacterial taxa in the gut microbiota, the number of piglets required in each group is 32. Based on previous observations, we have considered that 30 % of piglets will develop postweaning diarrhea indicating that 106 piglets would be required to obtain 32 piglets that develop postweaning diarrhea. To consider the variability in the occurrence of postweaning diarrhea we included 116 in our study. We also paid attention to collect samples from different litters (n=30 sows) since this factor strongly influences the gut microbiota in early life.

Figure 3 shows that the total family levels do not reach 100%. I would like to know the reason for this discrepancy.

The total family levels do not reach 100% since figure 3B represents the relative abundance of the 10 most abundant bacterial families (Figure legends, line 476).

The authors aim to find the effects of trimethylamine and butyrate on gene expression involved in proliferation and differentiation in piglet colon organoids using qPCR. However, it is not acceptable to rely solely on qPCR data, and the authors should also determine the protein levels to provide a more complete picture.

In a first attempt to answer to the reviewer request, we have tried to quantify the protein level of the cytokine CXCL8 secreted by organoids in the culture medium by using ELISA with pig specific antibodies (Biorad #AHP2392). However, the concentration of CXCL8 in the culture medium of organoids was below the detection threshold in all groups and we were thus not able to determine the effect of trimethylamine and butyrate on the secretion of this cytokine.

Then, following the reviewer comment, we performed a new organoid culture experiment in order to evaluate the effect of trimethylamine and butyrate on the epithelial barrier function without relying on qPCR. For that, we cultured monolayers of cells derived from pig colon organoid in order to measure the transepithelial electrical resistance (TEER), an indicator of epithelial paracellular permeability. We found that trimethylamine (1 mM) had no effect on TEER while butyrate (1 mM) increased TEER. Additionally, confocal microscopy imaging showed that TMA did not change the morphology of the epithelial cell monolayers nor on the expression of the tight junction protein occludin. These new results confirm our qPCR data showing that TMA do not alter colon epithelium homeostasis while butyrate strengthen the epithelial barrier. We have added these new experiments in the manuscript (methods: lines 406-416, results: lines 194-199, discussion: lines 282-286, legends: 512-521). Christelle Marraud was added as an author of the manuscript since she performed these new experiments.

Reviewer #2 (Comments for the Author):

Line 61-62: the format of reference is not correct;

We have corrected the format of the references.

Line 99-100: please make it clear the number of piglets in each room;

We have added the number of piglets in each room (lines 321-323).

Line 100-103: although there are different diarrhea scoring systems, all of them include light diarrhea and severe diarrhea, the difference on diarrhea may influence the microbiota a lot. However, current study only record diarrhea or not, and even include piglets only observed at once (these piglets may even not show microbiota changes), I will treat this as a big limitation of this study, this may cause the current results that no difference between normal and diarrhea piglets. If I can suggest, please use diarrhea samples that piglet show severe diarrhea in at least continue 3 days;

We used a binary score of diarrhea (yes/no) since we considered that it was the best way to identify piglets that always remained healthy after weaning versus the piglets that experienced mild or severe diarrhea, both subtypes being difficult to discriminate by scoring methods. We agree with the reviewer that piglets that we classified with postweaning diarrhea might have experienced mild or severe symptoms during a variable period. Yet, subclassification of piglets according to these criteria would have reduced the sample size, which is a strength of our study (n=37 piglets with postweaning diarrhea). Indeed, previous studies showing a link between the early life microbiota and the later development of postweaning diarrhea probably lacked statistical power (n=5 piglets with diarrhea in Dou et al., Plos One 2017). Moreover, the two other studies testing the capacity of the gut microbiota to predict postweaning diarrhea also used a binary score and did not considered the severity/duration of diarrhea (Dou et al, Plos One, 2018, PMID: 28072880 ; Karasova et al., Res Vet Sci, 2021, PMID: 33444908).

Following the comment of the reviewer, we have added this limitation of our study:

“The lack of identification of a clear microbiota signature in suckling piglets with later PWD could also be linked to the multiple possible causes of diarrhea after weaning (osmotic, secretory, inflammatory) (2). We cannot exclude that a gut microbiota signature might exist in some subtype of PWD and the lack of identification of the cause of diarrhea is a limitation of our study (e.g. by screening the presence of pathogens in the feces of piglets with PWD). Moreover, we have used a binary scoring of PWD (yes/no) to ensure a clear identification of piglets that remained healthy after weaning but this method did not allow us to distinguish piglets with mild or severe diarrhea neither according to the duration of symptoms. The large time scale of PWD development in our study (between postnatal day 23 and 54) might also induce variability due to different mechanisms potentially involved in early and late PWD. However, we did not perform sub-classes analysis in order to keep a high statistical power (n=37 piglets with PWD) which might have been lacking in previous studies (17).” (Lines 238-252)

Line 101-102: as far as I understand, feces samples will be collect at a very long period (1 month period: d21 to d 54), this means some samples may collected from d 22, dome samples may collected from d 50, this is the second shortage of this study: long time period will influence the results a lot! It means this will cause high variation to the results in the same group.

Fecal samples were collected only once for microbiota and metabolome analysis at postnatal day 13 (i.e. during the suckling period), as written lines 326-328. Thus, all samples used for microbiota and metabolome analyses were obtained from piglets at the same age. However, scoring of diarrhea was performed daily for each piglet from the weaning day (postnatal day 21) until day 54, as written lines 323-324. We agree with the reviewer that the long time period during which postweaning diarrhea occurred contribute to increase variability which might be a limitation of our study, as we have commented in the discussion “The large time scale of PWD development in our study (between postnatal day 23 and 54) might also induce variability due to different mechanisms potentially involved in early and late PWD. However, we did not perform sub-classes analysis in order to keep a high statistical power (n=37 piglets with PWD) which might have been lacking in previous studies (17)” (lines 238-252).

In order to further address the reviewer comment, we have reanalyzed our data by displaying the age at which diarrhea was recorded for the first time in each piglet on the nMDS that represents the microbiota structure (β -diversity). The results show that the samples collected from piglets that developed late diarrhea (eg. at day 37, 42, 43, 54) did not cluster separately from piglets that developed early diarrhea (eg. at day 26-30). Thus, we consider that the large time-scale of diarrhea occurrence in our study does not influence our conclusion that the early life microbiota is not a major factor underlying the susceptibility to post-weaning diarrhea. We have mentioned this observation in the discussion (Lines 238-252).

Line 104-105: not clear enough to show sample collection. In current description: "Fecal samples were collected during the suckling period at postnatal day 13 in healthy piglets...", do this mean only 1 time point of feces sample was collect?

Indeed, fecal samples were collected at a single time point during the suckling period at postnatal day 13. We have modified the text to make it clearer: “Fecal samples were collected once at postnatal day 13 in healthy piglets (i.e. during the suckling period) at postnatal day 13 in healthy piglets and stored at -80°C until analysis.” (lines 327-328)

Line 176-179: please make it clear in here what are the genes you have analyzed in this study.

We have added a description of the genes we have analyzed in this study. “The genes which expression was analyzed were selected to characterize the intestinal epithelium renewal and

differentiation and to evaluate the barrier function” (lines 400-402). The list of genes and the corresponding primers are shown in supplemental table 2.

Line 255 & 285: Please aware this is the results part, not discussion and reference should in this part.

We have removed all references from the results part.

Line 280-285: this should be in the method part other than in the results part;

All the details regarding these experiments in organoids are presented in the materials and methods section (lines 378-416). However, we believe that the reader will be helped by this short reminder of what was done since this part of the result sections correspond to a second *in vitro* experiment following the *in vivo* experiment that was presented above.

June 9, 2023

Dr. Martin Beaumont
Institut national de recherche pour l'agriculture l'alimentation et l'environnement
Castanet Tolosan
France

Re: Spectrum00694-23R1 (The early life microbiota is not a major factor underlying the susceptibility to postweaning diarrhea in piglets)

Dear Dr. Martin Beaumont:

Your manuscript has been accepted, and I am forwarding it to the ASM Journals Department for publication. You will be notified when your proofs are ready to be viewed.

Sincerely,

Maristela Camargo
Editor, Microbiology Spectrum
